# Is *COL1A1* Gene rs1107946 Polymorphism Associated with Sport Climbing Status and Flexibility?

**DOI:** 10.3390/genes13030403

**Published:** 2022-02-23

**Authors:** Mika Saito, Michał Ginszt, Ekaterina A. Semenova, Myosotis Massidda, Kinga Huminska-Lisowska, Monika Michałowska-Sawczyn, Hiroki Homma, Paweł Cięszczyk, Takanobu Okamoto, Andrey K. Larin, Edward V. Generozov, Piotr Majcher, Koichi Nakazato, Ildus I. Ahmetov, Naoki Kikuchi

**Affiliations:** 1Graduate School of Health and Sport Science, Nippon Sport Science University, Tokyo 158-8508, Japan; sa123ka.v@gmail.com (M.S.); hiroki0145@gmail.com (H.H.); tokamoto@nittai.ac.jp (T.O.); nakazato@nittai.ac.jp (K.N.); 2Department of Rehabilitation and Physiotherapy, Medical University of Lublin, 20-059 Lublin, Poland; michalginszt@umlub.pl (M.G.); piotr.majcher@umlub.pl (P.M.); 3Department of Molecular Biology and Genetics, Federal Research and Clinical Center of Physical-Chemical Medicine of the Federal Medical Biological Agency, 119435 Moscow, Russia; alisacemenova@mail.ru (E.A.S.); zelaz@yandex.ru (A.K.L.); generozov@gmail.com (E.V.G.); genoterra@mail.ru (I.I.A.); 4Research Institute of Physical Culture and Sport, Volga Region State University of Physical Culture, Sport and Tourism, 420010 Kazan, Russia; 5Department of Life and Environmental Sciences, University of Cagliari, 09124 Cagliari, Italy; myosotis.massidda@gmail.com; 6Faculty of Medicine and Surgery, Graduate School of Exercise and Sports Sciences, University of Cagliari, 09124 Cagliari, Italy; 7Faculty of Physical Education, Gdańsk University of Physical Education and Sport, 80-854 Gdańsk, Poland; kinga.huminska-lisowska@awf.gda.pl (K.H.-L.); monika.michalowska-sawczyn@awf.gda.pl (M.M.-S.); cieszczyk@poczta.onet.pl (P.C.); 8Department of Physical Education, Plekhanov Russian University of Economics, 115093 Moscow, Russia; 9Laboratory of Molecular Genetics, Kazan State Medical University, 420012 Kazan, Russia; 10Research Institute for Sport and Exercise Sciences, Liverpool John Moore University, Liverpool L3 5UX, UK

**Keywords:** genotype, heterozygosity, polymorphism, *COL1A1*, aging, flexibility, sit-and-reach, sports climbing

## Abstract

The purpose of this study was to compare the frequency of *COL1A1* rs1107946 polymorphism between sport climbers and controls from three ethnic groups (Japanese, Polish, and Russian) and investigate the effect of the *COL1A1* rs1107946 polymorphism on the age-related decrease in flexibility in the general population. Study I consisted of 1929 healthy people (controls) and 218 climbers, including Japanese, Polish, and Russian participants. The results of the meta-analysis showed that the frequency of the AC genotype was higher in climbers than in the controls (*p* = 0.03). Study II involved 1093 healthy Japanese individuals (435 men and 658 women). Flexibility was assessed using a sit-and-reach test. There was a tendency towards association between sit-and-reach and the *COL1A1* rs1107946 polymorphism (genotype: *p* = 0.034; dominant: *p* = 0.435; recessive: *p* = 0.035; over-dominant: *p* = 0.026). In addition, there was a higher negative correlation between sit-and-reach and age in the AA + CC genotype than in the AC genotype (AA + CC: r = −0.216, *p* < 0.001; AC: r = −0.089, *p* = 0.04; interaction *p* = 0.037). However, none of these results survived correction for multiple testing. Further studies are warranted to investigate the association between the *COL1A1* gene variation and exercise-related phenotypes.

## 1. Introduction

Sport climbing has become increasingly popular and was selected for inclusion in the 2020 Olympic Games. In particular, bouldering and lead climbing are performed professionally by athletes and as recreational activities by the general population. Climbing performance is associated with various physical and morphological characteristics. Previous studies have suggested that climbing performance is correlated with endurance performance, flexibility, and grip strength relative to body mass [1]. In addition, elite climbers have significantly lower fat mass percentage [2,3,4] and a higher grip strength relative to their body mass and flexibility when compared with controls and non-elite climbers [5,6]. These physical and morphological characteristics are affected by environmental and genetic factors. Furthermore, the heritability of athletic performance was estimated to be 66% in a twin study [7]. Saito, et al. [8] reported that the frequency of the X allele in *ACTN3* R577X polymorphism, which is associated with high flexibility [9], was higher in climbers than in controls.

Flexibility is influenced by both genetic and environmental factors. Previous studies on monozygotic and dizygotic twins suggest that 18–55% of the heritability in flexibility (based on assessments of “sit-and-reach”) is explained by genetic factors [10,11,12]. Gene polymorphisms associated with flexibility include the range of motion, muscle stiffness, and general joint laxity [9,13,14,15,16]. A recent study showed that *COL1A1* gene rs1107946 polymorphism is associated with muscle stiffness and injury [13]. Flexibility is influenced by the properties of intramuscular tissues [17]. Among the 28 identified types, type I collagen is an important component that affects intramuscular connective tissues properties [18], such as hysteresis, elasticity, and viscosity [19]. Therefore, elongation of the muscle tissue across the joint may affect the range of motion measured by the sit-and-reach test. A previous study suggested that the C allele in the *COL1A1* rs1107946 polymorphism tends to have a larger type I collagen α1/α2 chain ratio than the AA genotype, and lower muscle stiffness was shown in the C allele compared with the AA genotype [13]. Hence, *COL1A1* gene rs1107946 polymorphism may also be associated with the range of motion, but this remains to be verified.

Flexibility gradually decreases with age [20] and is affected by factors such as physical training and stretching [21,22,23,24]. The mechanical alteration of collagen molecules, which is important in the extracellular matrix, may affect the age-related decrease in flexibility. Pro-collagen I genes in the muscle are expressed in lower levels in older men than in younger men [25]. In addition, collagen turnover decreases in mature mice and induces fibrosis [26]. Therefore, age-related decrease in flexibility may be affected by genetic factors, such as *COL1A1* rs1107946 polymorphism. In addition, continuous training and stretching to improve athletic performance are adapted to athlete-specific physical fitness. For example, elite climbers have higher flexibility than recreational climbers [6,27]. Therefore, the climbers’ status and flexibility in general populations may be influenced by *COL1A1* rs1107946 polymorphism.

The purpose of this study was (1) to compare the frequency of *COL1A1* rs1107946 polymorphism between climbers and controls in three ethnic groups (Japanese, Polish, and Russian), and (2) to investigate the effect of *COL1A1* rs1107946 polymorphism on the age-related decrease in flexibility in the general population.

## 2. Materials and Methods

### 2.1. Study I

#### 2.1.1. Subjects

Study I consisted of 1929 healthy people (controls) and 218 climbers. The Japanese cohort included 1093 controls (*n* = 435 males, *n* = 658 females; age 54.3 ± 15.2 years) and 62 climbers (38 males, 24 females; age 23.2 ± 8.4 years). The Polish cohort included 635 controls (*n* = 505 males, *n* = 130 females; age 25.1 ± 5.0 years) and 126 climbers (97 males, 29 females; age 28.4 ± 6.1 years). The Russian cohort included 201 controls (*n* = 158 males, *n* = 43 females; age 45 ± 4.3 years) and 30 climbers (17 men, 13 women; age 22.8 ± 4.1 years). All participants were informed of the purpose and methods of the study, and each provided written informed consent for participation. This study was approved by the ethics committees of the Nippon Sport Science University, Gdańsk University of Physical Education and Sport, and the Federal Research and Clinical Center of Physical-Chemical Medicine of Federal Medical Biological Agency. This study was conducted in accordance with the principles of the Declaration of Helsinki for Human Research.

#### 2.1.2. Genotyping

##### Japanese Climbers and Controls and Polish Climbers

Total DNA was extracted and isolated from the saliva of the participants using an Oragene-DNA kit (DNA Genotek, ON, Canada). The *COL1A1* rs1107946 polymorphism was genotyped using the TaqMan SNP Genotyping Assay (Assay ID: *COL1A1* gene rs1107946 polymorphisms: C___7477171_10) using a Bio-Rad PCR System (CFD-3120J1, Bio-Rad, Hercules, CA, USA). The genotyping mixture (total volume: 5 μL) contained 2.5 μL of GTXpress Master Mix, 0.125 μL of assay mix (40×), and 1.375 μL of distilled water, with 1 μL of genomic DNA (10 ng/μL) per reaction. The thermal cycling conditions included an initial denaturation at 95 °C for 20 s, followed by 40 cycles of denaturation at 95 °C for 3 s, and annealing/extension at 60 °C for 20 s.

##### Polish Controls

To obtain the DNA of *COL1A1* rs1107946 polymorphisms from the Polish controls, samples were extracted from the buccal cells using a High Pure PCR Template Preparation Kit (Roche, Switzerland) according to the manufacturer’s instructions. The genotyping mixture (total volume: 5 μL) contained 2.5 μL of TaqPath ProAmp Master Mix (Thermo Fisher Scientific, Waltham, MA, USA), 0.25 μL of assay mix (10×), and 1 μL of distilled water with 1.25 μL of genomic DNA (~10 ng/μL) per reaction. The thermal cycling conditions included a pre-read at 60 °C for 30 s, an initial denaturation at 95 °C for 5 min, followed by 40 cycles of denaturation at 95 °C for 5 s, annealing/extension at 60 °C for 30 s, and a post-read at 60 °C for 30 s. Genotyping reactions were performed on a CFX Connect Real-Time PCR Detection System (BioRad, Hercules, CA, USA).

##### Russian Climbers and Controls

In Russian climbers and controls, molecular genetic analysis was performed using DNA samples obtained from leukocytes (venous blood). Venous blood samples (4 mL) were collected in tubes containing ethylenediaminetetraacetic acid (EDTA) (Vacuette EDTA tubes, Greiner Bio-One, Kremsmünster, Austria). Blood samples were transported to the laboratory at 4 °C, and the DNA was extracted on the same day. DNA extraction and purification were performed using a commercial kit (Techno-Sorb, Moscow, Russia) according to the manufacturer’s instructions (Technoclon, Moscow, Russia). The protocol included chemical lysis, selective DNA binding on silica spin columns, and ethanol washing. The quality of the extracted DNA was assessed using agarose gel electrophoresis. HumanOmni1-Quad BeadChips (Illumina Inc., San Diego, CA, USA) were used for genotyping 1,140,419 single-nucleotide polymorphisms (SNPs; including COL1A1 polymorphism) in athletes and controls. The assay required 200 ng of DNA sample input at a concentration of at least 50 ng/µL. The exact concentrations of DNA in each sample were measured using a Qubit Fluorometer (Invitrogen, Waltham, Massachusetts, USA). All further procedures were performed according to the instructions provided for the Infinium HD Assay (Illumina, San Diego, CA, USA).

### 2.2. Study II

#### 2.2.1. Subjects

Study II included 1093 healthy Japanese individuals (435 men, age: 52.7 ± 15.7 years, height: 170.4 ± 6.4 cm, weight: 69.3 ± 10.6 kg, and 658 women: age: 55.4 ± 14.7 years, height: 156.7 ± 5.7 cm, weight: 54.1 ± 8.2 kg). Flexibility was measured using a sit-and-reach test (in cm) with a digital measuring device (Takei Scientific Instruments Co. Ltd., Tokyo, Japan). The subjects were instructed to sit on the floor and stretch their legs out in front of them with their knees straight. The subjects were allowed two to three attempts to attain the highest possible rating. This study was approved by the Ethics Committee of Nippon Sport Science University. This study was conducted in accordance with the principles of the Declaration of Helsinki for Human Research.

#### 2.2.2. Genotyping

We used the same protocol as in study I of Japanese controls.

### 2.3. Statistical Analysis

The SPSS statistical package version 25.0 for Mac was used to perform all statistical analyses. Genotype and allele frequencies were calculated for all gene polymorphisms, and Hardy–Weinberg equilibrium was assessed using the chi-square (×2) test. In addition, a meta-analysis of the three ethnic cohorts was conducted to investigate the association between the frequency of each gene polymorphism in climbers and controls. The meta-analysis was conducted using the Review Manager software program (version 5.3; Copenhagen: The Nordic Cochrane Center, The Cochrane Collaboration) [28]. Random effect models were applied. Odds ratio with 95% confidence intervals (CI) was estimated using the Mantel–Haenszel method. The heterogeneity degree between the studies was assessed with the *I*^2^ statistics.

Age-related decrease in sit-and-reach was examined using correlation analysis. The association between *COL1A1* gene rs1107946 polymorphism and age, height, and weight were examined using one-way analysis of variance (ANOVA). In addition, the association between *COL1A1* gene rs1107946 polymorphism and sit-and-reach was examined by one-way analysis of covariance (ANCOVA), adjusting for sex and age. Age-related effects on the association between *COL1A1* gene rs1107946 polymorphism and sit-and-reach were assessed from the difference in the r-value (slope of the line) of the correlation between age and sit-and-reach, and two-way ANCOVA was adjusted for sex. A Bonferroni adjustment for multiple testing (climbing status and flexibility) was made with statistical significance set at *p* < 0.025.

## 3. Results

### 3.1. Study I

The *COL1A1* rs1107946 frequencies among climbers and controls from Poland and Russia were in Hardy–Weinberg equilibrium (Polish: climbers *p* = 0.383, controls *p* = 0.868; Russian: climbers *p* = 0.461, controls *p* = 0.090), but not in Japanese climbers (*p* = 0.035). There were no significant differences in the distribution of *COL1A1* rs1107946 between the climbers and controls in any ethnic cohort for genotype and allele models (Table 1). In addition, the odds ratio of each genetic model (dominant, recessive, and additive) was not significantly associated with any ethnic cohort (Table 2). There was no significant sex difference of genotype frequency in Japanese (*p* = 0.657), Polish (*p* = 0.721), and Russian (0.549).

The results of the meta-analysis across the three cohorts showed a trend towards significance of AC genotype frequency in climbers compared with AA and CC genotype (Figure 1). There were no associations in the dominant and recessive models. The frequency of the AC genotypes in the *COL1A1* rs1107946 polymorphism was higher in Japanese, Polish, and Russian climbers than in the controls (Figure 1 odds ratio: 1.41, 95% confidence interval (CI):1.03–1.93, *p* = 0.03). No heterogeneity between studies (*I*^2^ = 0%; *p* = 0.98) was observed.

### 3.2. Study II

The frequencies of *COL1A1* rs1107946 polymorphism among the Japanese controls were in Hardy–Weinberg equilibrium (*p* = 0.389). We show the associations between *COL1A1* rs1107946 polymorphism and sit-and-reach in Table 3. There was a trend of associations between sit-and-reach and *COL1A1* rs1107946 polymorphism in genotype, recessive, and over-dominant models. A trend towards significance in sit-and-reach was observed when CC + AC were compared with AA genotype (recessive model: *p* = 0.035). In addition, the AC genotype showed higher flexibility than the AA + CC genotype (over-dominant model: *p* = 0.026). There was significant negative correlation between sit-and-reach and age (r = −0.151, *p* < 0.001), with an interaction effect between them when the AA + CC genotype group was compared with the AC genotype (AA + CC: r = 0.21, *p* = 2.35 × 10^–7^, AC: r = −0.09, *p* = 0.04; interaction *p* = 0.037; Figure 2) but not in other the models.

## 4. Discussion

One of the purposes of this study was to examine the frequency of COL1A1 rs1107946 polymorphism in climbers, who require higher flexibility and strength, from three ethnicities (Study I). The results of the meta-analysis across the three ethnicities (Japanese, Polish, and Russian) showed that the frequency of the AC genotype was higher in climbers than in the controls (*p* = 0.03). In addition, our results showed a trend towards significance between the *COL1A1* rs1107946 C allele (AC and CC genotype) and flexibility in the general population (Study II). There was a tendency of interaction effect of correlation between sit-and-reach and age in the AC genotype and the AA + CC genotype (AA + CC: r = −0.216, *p* < 0.001; AC: r = −0.089, *p* = 0.04; interaction *p* = 0.037). These results suggest that the AC genotype may have an advantage over the age-related decrease in flexibility, such as sit-and-reach. However, none of these results survived correction for multiple testing (*p* < 0.025).

In the present study, our meta-analysis across three ethnicities (Japanese, Polish, and Russian) shows that the frequency of the AC genotype is higher in climbers than in controls (*p* = 0.03). Flexibility is an important component of climbing performance. Sit-and-reach tests were positively correlated with climbing performance [1]. Studies have suggested that sit-and-reach is higher in elite climbers than in recreational climbers [6,27]. In addition, climbing-specific flexibility, such as foot raises in front of the wall, is higher in elite climbers than recreational- and non-climbers [5]. These flexibilities may be influenced by muscle strength. Collectively, our results suggest that climbers require both the range of motion and the muscle strength to control it. Therefore, the AC genotype in *COL1A1* might be related to not only flexibility, but also sport climbing performance.

A previous study reported an association between COL1A1 rs1107946 polymorphism and flexibility (e.g., muscle stiffness) [13]. The study suggested that the C allele (CC + AC genotype) of COL1A1 rs1107946 polymorphism shows lower muscle stiffness than the AA genotype. Our results are consistent with this finding. In addition, we observed an age-related decrease in flexibility (r = −0.151, *p* < 0.001), which was the same in men (r = 0.186, *p* < 0.001) and women (r = −0.156, *p* < 0.001). Physical ability, including flexibility and muscle strength, is impaired with age [20]. The collagen turnover decreases with aging [26], which increases the non-enzymatic cross-linking of collagen fibers through the accumulation of advanced glycation end-products [29]. Previous studies have suggested that the fibrotic phenotype increases muscle stiffness [30]. The fibrotic collagen phenotype may be prevented by the higher COL1A1 mRNA expression, promoting the turnover of collagens. A previous study suggested that age-related decrease in muscle strength and muscle cross-sectional area may be suppressed by genetic factors [31,32]. Our study showed a higher age-related decrease in the AA + CC genotype compared to the AC genotype in COL1A1 rs1107946 polymorphism. Thus, the AC genotype in COL1A1 rs1107946 polymorphism may weaken the age-related decrease in flexibility.

The sit-and-reach test in our study may reflect muscle and musculotendinous stiffness and muscle strength. Furthermore, muscle stiffness and strength also exhibit a trade-off relationship. Alfuraih, et al. [33] suggested that muscle stiffness measured in the lower limb muscles is positively correlated with isokinetic knee strength. A previous study on the osteogenesis imperfecta murine (oim) model suggested that oim/oim mice exclusively produce homotrimeric (three α1 chains) type I collagen and had decreased tetanic force and body weight compared to the +/oim genotypes [34]. Higher levels of type I collagen may reduce muscle strength. In contrast, the C allele of COL1A1 rs1107946 polymorphism results in a higher collagen type I α1/α2 ratio and lower muscle stiffness [13]. Therefore, muscle strength and stiffness, which influence sit-and-reach, may be differently affected by the COL1A1 rs1107946 polymorphism.

The athletic ability that requires both flexibility and muscle strength, such as sit-and-reach and climbing-specific flexibility, may require both genetic characteristics of *COL1A1* rs1107946 gene polymorphism. Heterozygosity is a measure of genetic diversity and may have an effect on the phenotypes [35,36]. For example, muscle injury and decreased bone mineral density (BMD) have opposing relationships with the *COL1A1* rs1107946 polymorphism. Miyamoto-Mikami, et al. [13] reported that athletes with muscle injury have a higher frequency of the AA + AC genotype in *COL1A1* rs1107946 polymorphism compared to the CC genotype. In contrast, a previous study suggested that the CC genotype carriers had significantly lower lumbar spine BMD than AA genotype carriers [13]. The CC genotype in *COL1A1* rs1107946 polymorphism may be a risk factor for low BMD and fatigue fractures [13,37,38,39]. Therefore, there is a trade-off between muscle and bone injury: the A allele in *COL1A1* is associated with higher BMD, while the C allele decreases the risk of muscle injury.

The AC genotype has a heterozygous effect on complex phenotypes, such as sit-and-reach and climbing status, which require flexibility and muscle strength. However, the function of the AC genotype in *COL1A1* rs1107946 remains unclear and would require more functional and longitudinal studies in the future.

## 5. Conclusions

In our meta-analysis of three ethnicities, sports climbers, who require higher flexibility and muscle strength, have a higher AC genotype frequency than control participants. In addition, the C allele, especially in AC genotype, might be related to flexibility in general populations. However, none of these results survived correction for multiple testing. Further studies are warranted to investigate the association between the COL1A1 gene variation and exercise-related phenotypes.

## Figures and Tables

**Figure 1 genes-13-00403-f001:**
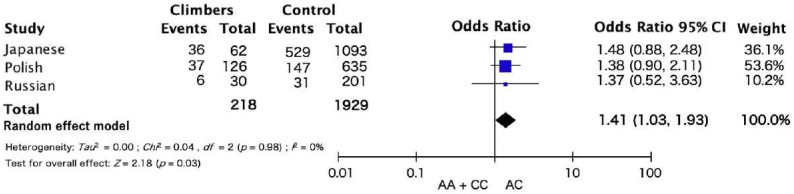
Meta-analysis results of three ethnic cohorts for the association *COL1A1* rs1107946 polymorphism and climbing status.

**Figure 2 genes-13-00403-f002:**
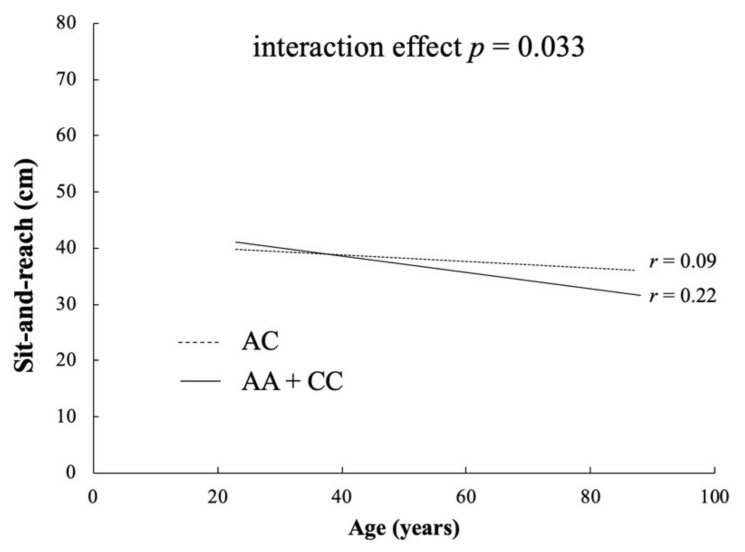
Correlation between sit-and-reach and age for the AC and AA + CC genotypes in *COL1A1* rs1107946 polymorphism.

**Table 1 genes-13-00403-t001:** Frequency of genotype and allele model of *COL1A1* rs1107946 polymorphism in Japanese, Polish, and Russian climbers and controls.

		Genotype			*p*-Value	Allele		*p*-Value
		AA	AC	CC		A	C	
Japanese								
	Climbers	4 (6)	36 (58)	22 (36)	0.164	44 (35)	80 (65)	0.558
	Controls	152 (14)	529 (48)	412 (38)		833 (38)	1353 (62)	
Polish								
	Climbers	2 (2)	37 (29)	87 (69)	0.328	41 (16)	211 (84)	0.240
	Controls	12 (2)	147 (23)	476 (75)		171 (13)	1099 (87)	
Russian								
	Climbers	1 (3)	6 (20)	23 (77)	0.715	8 (13)	52 (87)	0.385
	Controls	4 (2)	31 (15)	166 (83)		39 (10)	363 (90)	

**Table 2 genes-13-00403-t002:** Odds ratio of each genetic model (dominant, recessive, and additive) in Japanese, Polish, and Russian climbers.

		Japanese				Polish			Russian	
		OR	95%CI	*p*-Value	OR	95%CI	*p*-Value	OR	95%CI	*p*-Value
Dominant										
	CC	1.00		0.73	1.00		0.17	1.00		0.45
	AA + AC	0.91	0.53–1.55		0.75	0.49–1.13		0.69	0.28–1.74	
Recessive										
	CC + AC	1.00		0.068	1.00		0.81	1.00		0.66
	AA	2.34	0.84–6.54		1.19	0.26–5.40		0.59	0.06–5.45	
Over dominant										
	AA + CC	1.00		0.14	1.00		0.14	1.00		0.53
	AC	0.68	0.40–1.14		0.72	0.47–1.11		0.73	0.28–1.93	

**Table 3 genes-13-00403-t003:** Association between *COL1A1* rs1107946 and sit and reach.

		*n*	Age	Height	Weight	Sit-and-Reach
Genotype	AA	152	55.3 ± 15.3	161.2 ± 8.2	60.1 ± 10.1	35.6 ± 10.2
	CA	529	55.1 ± 15.2	161.9 ± 9.4	59.7 ± 12.1	37.9 ± 10.1
	CC	412	53.0 ± 15.1	162.7 ± 8.7	60.8 ± 12.1	37.0 ± 10.1
Dominant	AA + AC	681	55.1 ± 15.2	161.7 ± 9.1	59.8 ± 11.7	37.4 ± 10.2
Recessive	CC + AC	941	54.1 ± 15.1	162.3 ± 9.1	60.2 ± 12.1	37.5 ± 10.1
Over dominant	AA + CC	564	53.6 ± 15.2	162.3 ± 8.6	60.6 ± 11.6	36.6 ± 10.1
*p*-value	Genotype		0.071	0.152	0.33	0.034 *
	Dominant		0.022	0.076	0.149	0.435 *
	Recessive		0.375	0.194	0.919	0.035 *
	Over dominant		0.108	0.412	0.184	0.026 *

Mean ± SD. * adjusted by age and sex.

## Data Availability

The data presented in this study are available on request from the corresponding author.

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
