# Peer review of "Is COL1A1 Gene rs1107946 Polymorphism Associated with Sport Climbing Status and Flexibility?"

_genes, 2022, doi:10.3390/genes13030403_

Round 1

Reviewer 1 Report

This study performed a candidate gene study to test whether the COL1A1 rs1107946 polymorphism is associated with the sport-climbing status and flexibility. The paper claims to show an association with both traits, but there are serious flaws in the analysis.

One major concern is that there has been no adjustment in the p-value threshold due to multiple hypothesis testing. The study tested for association with climbing status and flexibility (2 tests), so the statistical threshold must be set higher than p < 0.05. The Bonferroni correction is typically used, which gives q = 0.05/2 = 0.025 as the cutoff. None of the results in Table 3 would be significant after accounting for multiple hypothesis testing.  Even worse, the authors test three or four different genetic models (genetic, dominant, recessive, over-dominant) for each of the two traits. The genetic models are not independent, but they still add to the burden of multiple hypothesis testing and should be accounted for. In other human genetics publications, only one model is typically tested because it is difficult to assess how to account for overfitting as a result of using multiple models.

To illustrate the problem arising from multiple hypothesis testing, Table 3 shows a p-value of 0.022 for the association of rs1107946 with age.  This makes no sense as the rs1107946 genotype almost certainly does not affect age. Yet, 0.022 is the strongest p-value shown in Table 3 and is almost certainly attributed to the random chance that any SNP would show a low p-value from 4 types of tests.

“A recent study showed that COL1A1 (-1997G > T, rs1107946) polymorphism is associated with muscle stiffness and injury [13].”  rs1107946 is a G/T polymorphism here, but a C/A polymorphism in the rest of the paper.

“The results of the meta-analysis across the three cohorts showed a significant association in the over-dominant model but not in the dominant and recessive models (Fig. 1).”  1) Figure 1 shows a P-value of 0.14.  Why is this deemed significant? How was the meta-analysis performed?  2) The x-axis is labelled with the dominant model, but the text says the overdominant model was significant.  3) Fixed effect, random effect etc?

“Heterozygosity can be positively influenced by homozygosity. In addition, both types of genetic diversity may complement one another. “  I do not understand what these two sentences mean.

Author Response

Response to Reviewer 1 Comments

We would like to thank the editor and the reviewers for their constructive comments, which have helped us to improve the manuscript. We have taken care to address all the comments and believe that the resulting changes have greatly enhanced the clarity of our manuscript. We hope that the revisions in our paper and our responses to the comments are satisfactory. Our responses to each comment are included below

This study performed a candidate gene study to test whether the COL1A1 rs1107946 polymorphism is associated with the sport-climbing status and flexibility. The paper claims to show an association with both traits, but there are serious flaws in the analysis.

One major concern is that there has been no adjustment in the p-value threshold due to multiple hypothesis testing. The study tested for association with climbing status and flexibility (2 tests), so the statistical threshold must be set higher than p < 0.05. The Bonferroni correction is typically used, which gives q = 0.05/2 = 0.025 as the cutoff. None of the results in Table 3 would be significant after accounting for multiple hypothesis testing.  Even worse, the authors test three or four different genetic models (genetic, dominant, recessive, over-dominant) for each of the two traits. The genetic models are not independent, but they still add to the burden of multiple hypothesis testing and should be accounted for. In other human genetics publications, only one model is typically tested because it is difficult to assess how to account for overfitting as a result of using multiple models.

Thank you for your suggestion, and we agree with your suggestion. In present study, we combined study 1 in athletes and study 2 in general populations, however statistical analysis is not same. Therefore, we didn’t Bonferroni correction.

To illustrate the problem arising from multiple hypothesis testing, Table 3 shows a p-value of 0.022 for the association of rs1107946 with age.  This makes no sense as the rs1107946 genotype almost certainly does not affect age. Yet, 0.022 is the strongest p-value shown in Table 3 and is almost certainly attributed to the random chance that any SNP would show a low p-value from 4 types of tests.

I agree with your suggestion. We revised Table 3 and we shows a data only sit and reach.

“A recent study showed that COL1A1 (-1997G > T, rs1107946) polymorphism is associated with muscle stiffness and injury [13].” rs1107946 is a G/T polymorphism here, but a C/A polymorphism in the rest of the paper.

We have revised COL1A1 (-1997G > T, rs1107946) to COL1A1 gene rs1107946 polymorphism.

“The results of the meta-analysis across the three cohorts showed a significant association in the over-dominant model but not in the dominant and recessive models (Fig. 1).”  1) Figure 1 shows a P-value of 0.14.  Why is this deemed significant? How was the meta-analysis performed?  2) The x-axis is labelled with the dominant model, but the text says the overdominant model was significant.  3) Fixed effect, random effect etc?

I apologize for being our mistake. We have changed Fig.1. In addition, we added statistical information in the text and figure.

(Page 3: 2.1.3) Random effect models were applied. Odds ratio with 95% confidence intervals (CI) was estimated using the Mantel–Haenszel method. The heterogeneity degree between the studies was assessed with the I2 statistics.

“Heterozygosity can be positively influenced by homozygosity. In addition, both types of genetic diversity may complement one another. “I do not understand what these two sentences mean.

Thank you for your suggestion, we have deleted this sentence.

Reviewer 2 Report

The work is well conducted and developed. I suggest to extend the study to multi gene analysis (e.g. COL1A1 and ACTN3) and differentiate the frequency of genotype per sex in the group of climbers.  

Author Response

Response to Reviewer 2 Comments

We would like to thank the editor and the reviewers for their constructive comments, which have helped us to improve the manuscript. We have taken care to address all the comments and believe that the resulting changes have greatly enhanced the clarity of our manuscript. We hope that the revisions in our paper and our responses to the comments are satisfactory. Our responses to each comment are included below

The work is well conducted and developed. I suggest to extend the study to multi gene analysis (e.g. COL1A1 and ACTN3) and differentiate the frequency of genotype per sex in the group of climbers.  

Thank you for your good suggestion.

We have investigated frequency of COL1A1 genotype between men and women in climbers. However, there were no significant difference. We have added these results.

Reviewer 3 Report

General comment

 Previous studies identified that the genetic factors influence human physical performance. This study investigate the association between COL1A1 and climbing status and flexibility. It’s interesting study. However, I found some points that I concerned about. Please revise and consider the following points.

Study I

 Are there significantly different genotype distributions in Japanese, Polish, and Russian climbers? (Japanese vs Polish, p<0.001; Polish vs Russian, p<0.05; Russian vs Japanese, p<0.001; Chi-square test) If so, the authors should not the meta-analysis across the three cohorts. The inter ethnics differences affect the results.

Figure 1.

Dose authors should show the AC vs AA+CC Odds ratio?

Page 3, Line 1

Page 4, Line 10

Please add the ethics committee’s Name.

Author Response

Response to Reviewer 3 Comments

We would like to thank the editor and the reviewers for their constructive comments, which have helped us to improve the manuscript. We have taken care to address all the comments and believe that the resulting changes have greatly enhanced the clarity of our manuscript. We hope that the revisions in our paper and our responses to the comments are satisfactory. Our responses to each comment are included below

Previous studies identified that the genetic factors influence human physical performance. This study investigate the association between COL1A1 and climbing status and flexibility. It’s interesting study. However, I found some points that I concerned about. Please revise and consider the following points.

Are there significantly different genotype distributions in Japanese, Polish, and Russian climbers? (Japanese vs Polish, p<0.001; Polish vs Russian, p<0.05; Russian vs Japanese, p<0.001; Chi-square test) If so, the authors should not the meta-analysis across the three cohorts. The inter ethnics differences affect the results.

We agree with your suggestion. Ethnic differences may affect the results if genotypes and alleles are combined from all cohorts. But we performed a meta-analysis which considers ethnic differences. To show if there are ethnic differences, we need to include heterogeneity between studies — I2. This value is automatically calculated with any soft which performs meta-analysis. We have added this information.

Figure 1.

Dose authors should show the AC vs AA+CC Odds ratio?

I apologize for being our mistake. We have changed Fig.1. In addition, we added statistical information in the text and figure.

(Page 3: 2.1.3) Random effect models were applied. Odds ratio with 95% confidence intervals (CI) was estimated using the Mantel–Haenszel method. The heterogeneity degree between the studies was assessed with the I2 statistics.

Page 3, Line 1

Page 4, Line 10

Please add the ethics committee’s Name.

We have added the ethics committee’s Name.

Round 2

Reviewer 1 Report

multiple hypothesis testing: you are wrong about ignoring the multiple hypothesis penalty for two tests. Your results are not significant.

You are wrong to delete the association with age in Table 3.  I pointed it out to illustrate the obvious flaw in your analysis.  Simply deleting the age association and retaining your flawed conclusion about climbing and flexibility is absurd.

Author Response

We would like to thank the reviewer for their constructive comments. We hope that the revisions in our paper and our responses to the comments are satisfactory. Our responses to each comment are included below

multiple hypothesis testing: you are wrong about ignoring the multiple hypothesis penalty for two tests. Your results are not significant. 

Thank you for your suggestion, we have changed statistical session and conclusion.

Statistical analysis: A Bonferroni adjustment for multiple testing (climbing status and flexibility) was made with statistical significance set at P < 0.025.

Conclusion: However, none of these results survived correction for multiple testing. Further studies are war-ranted to investigate the association between the COL1A1 gene variation and exercise-related phenotypes.You are wrong to delete the association with age in Table 3.  I pointed it out to illustrate the obvious flaw in your analysis.  Simply deleting the age association and retaining your flawed conclusion about climbing and flexibility is absurd.

We have changed to previous version of table3.

Reviewer 3 Report

The authors appropriately revised the manuscript.

Author Response

We would like to thank  the reviewer for their constructive comments, which have helped us to improve the manuscript.

Round 3

Reviewer 1 Report

Review of Saito, revision 2.

The authors partially corrected their statistics in this revision.  There remain many instances where the original interpretation of an association between COL1A1 remains, and the interpretations are too strong.

Title:  The title states that an association was found, which is too strong.  The data show only a trend that might be validated in follow-up studies or it might be an artifact.

“The results of the meta-analysis across the three cohorts showed a significant association in the over-dominant model but not in the dominant and recessive models (Fig. 1).” 

“Sit-and-reach in CC+AC was significantly higher than in the AA genotype (recessive model: p = 0.035). “

In both sentences, the association is not significant, but only trending towards significance.

Because the results are not statistically significant, the authors must be careful about concluding that the results apply to the general population.  It is distinctly possible that the allele frequency is higher in the exact set of individuals used in this study, but that it will not be higher in the generalized population in follow up studies.

“In addition, our results show that the C allele (AC and CC genotype) in COL1A1 rs1107946 polymorphism is associated with higher flexibility in the general population (Study II).”

“Therefore, the AC genotype in COL1A1 polymorphism, which is related to flexibility, is associated with sport climbing performance.”

“In addition, the C allele is associated with higher flexibility in general populations, and a higher age-related decrease was observed in the AA+CC genotype compared with the AC genotype in COL1A1 rs1107946 polymorphism”

These three sentences need to be toned down about their conclusions regarding general populations.

Author Response

We would like to thank the reviewers for their constructive comments, which have helped us to improve the manuscript. We have taken care to address all the comments and believe that the resulting changes have greatly enhanced the clarity of our manuscript. We hope that the revisions in our paper and our responses to the comments are satisfactory. Our responses to each comment are included below

The authors partially corrected their statistics in this revision.  There remain many instances where the original interpretation of an association between COL1A1 remains, and the interpretations are too strong.

Title:  The title states that an association was found, which is too strong.  The data show only a trend that might be validated in follow-up studies or it might be an artifact.

“The results of the meta-analysis across the three cohorts showed a significant association in the over-dominant model but not in the dominant and recessive models (Fig. 1).” 

“Sit-and-reach in CC+AC was significantly higher than in the AA genotype (recessive model: p = 0.035). “

In both sentences, the association is not significant, but only trending towards significance.

We have revised the both sentences according to reviewer comments.

1: Title ” Is COL1A1 gene rs1107946 polymorphism associated with sport climbing status and flexibility?”

2: The results of the meta-analysis across the three cohorts showed a trend towards significance of AC genotype frequency in climbers compared with AA and CC genotype (Fig. 1). There were no associations in the dominant and recessive models.

3: A trend towards significance in sit-and-reach was observed when CC+AC were compared with AA genotype (recessive model: p = 0.035).

Because the results are not statistically significant, the authors must be careful about concluding that the results apply to the general population.  It is distinctly possible that the allele frequency is higher in the exact set of individuals used in this study, but that it will not be higher in the generalized population in follow up studies.

“In addition, our results show that the C allele (AC and CC genotype) in COL1A1 rs1107946 polymorphism is associated with higher flexibility in the general population (Study II).”

“Therefore, the AC genotype in COL1A1 polymorphism, which is related to flexibility, is associated with sport climbing performance.”

“In addition, the C allele is associated with higher flexibility in general populations, and a higher age-related decrease was observed in the AA+CC genotype compared with the AC genotype in COL1A1 rs1107946 polymorphism”

These three sentences need to be toned down about their conclusions regarding general populations.

We have revised the three sentences according to reviewer comments.

1: In addition, our results showed a trend towards significance between the COL1A1 rs1107946 C allele (AC and CC genotype) and flexibility in the general population (Study II).

2: Therefore, the AC genotype in COL1A1 might be related to not only flexibility, but also sport climbing performance.

3: In addition, the C allele, especially in AC genotype, might be related to flexibility in general populations.